# Merge-Friendly Post-Training Quantization for Multi-Target Domain Adaptation

**Juncheol Shin** [1]  **Minsang Seok** [2]  **Seonggon Kim** [2]  **Eunhyeok Park** [1][2]

## Abstract

Model merging has emerged as a powerful technique for combining task-specific weights, achieving superior performance in multi-target domain adaptation. However, when applied to practical scenarios, such as quantized models, new challenges arise. In practical scenarios, quantization is often applied to target-specific data, but this process restricts the domain of interest and introduces discretization effects, making model merging highly non-trivial. In this study, we analyze the impact of quantization on model merging through the lens of error barriers. Leveraging these insights, we propose a novel post-training quantization, HDRQ - Hessian and distant regularizing quantization - that is designed to consider model merging for multi-target domain adaptation. Our approach ensures that the quantization process incurs minimal deviation from the source pre-trained model while flattening the loss surface to facilitate smooth model merging. To our knowledge, this is the first study on this challenge, and extensive experiments confirm its effectiveness.

## 1. Introduction

Large-scale models have driven breakthroughs across various domains, particularly in generative AI, enabling efficient adaptation to multiple tasks and user-specific data. However, deploying such models on resource-constrained devices remains a significant challenge due to their computational demands. In this perspective, model merging has emerged as a promising technique that enables multi-target domain adaptation without additional training. A recent study on multi-

[1]Graduate School of Artificial Intelligence, Pohang University of Science and Technology (POSTECH), Pohang, Republic of Korea [2]Department of Computer Science and Engineering, Pohang University of Science and Technology (POSTECH), Pohang, Republic of Korea. Correspondence to: Eunhyeok Park <eh.park@postech.ac.kr>.

*Proceedings of the $42^{nd}$ International Conference on Machine Learning*, Vancouver, Canada. PMLR 267, 2025. Copyright 2025 by the author(s).

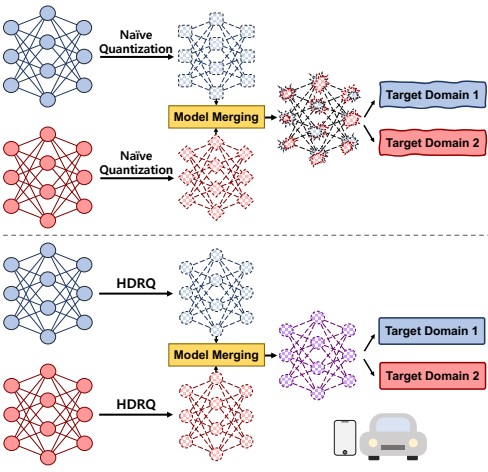

*Figure 1.* We propose a quantization scheme designed with future merging in mind. Our method ensures that networks are quantized to a more merge-friendly state, reducing the degradation induced by merging.

target domain adaptation (Li et al., 2024) demonstrated that models fine-tuned for different target domains can be fused into a single general model via simple weight averaging, even in a training-free manner. This discovery highlights the potential of real-time adaptive AI.

Despite its promise, a major obstacle in achieving practical multi-target domain adaptation is the effect of quantization. Quantization is essential for reducing memory and computational costs for efficiency, but it introduces discretization that is not well aligned with the merging idea, leading to sub-optimal merging and degraded performance. While previous works have extensively explored quantization and domain adaptation separately, little attention has been given to their interplay. In particular, no existing research systematically investigates how quantization influences model merging or proposes solutions to mitigate its impact.

To address this challenge, we introduce **HDRQ** (Hessian and Distance Regularizing Quantization), the first post-training quantization (PTQ) method designed to preserve merging compatibility in multi-target domain adaptation. Our key contributions are as follows:

- **Theoretical Analysis of Quantization's Impact on Model Merging:** We extend the concept of the error barrier (Frankle et al., 2020) to analyze how weight perturbations from quantization affect merging quality. Our study reveals that quantization-induced misalignment reduces merging effectiveness across different adaptation scenarios.

- **Regularization for Merge-Friendly Quantization:** Based on our analysis, HDRQ incorporates two key regularization techniques to ensure that quantized models remain merge-compatible:

    - *Hessian Regularization:* By controlling sensitivity to perturbations, we mitigate the adverse effects of quantization on merging stability.
    - *Distance Regularization:* By reducing weight divergence among quantized models, we enhance their ability to merge effectively.

- **Noise-Sampling-Based Rounding:** We introduce an advanced rounding mechanism that resolves the rounding ambiguity problem in conventional quantization, ensuring more stable weight updates.

We evaluate HDRQ across multiple datasets and compare it against conventional PTQ methods. Our key findings are:

- **Comparable or Superior Single-Model Performance:** HDRQ maintains accuracy on individual quantized models, performing on par with or better than existing PTQ methods.

- **Significantly Improved Merging Performance:** Unlike standard PTQ, which degrades merging quality, HDRQ ensures that merged models achieve higher accuracy and better generalization. For example, HDRQ improves the performance of merged model by 4.21 mIoU compared to conventional PTQ in the multi-target domain adaptation task for semantic segmentation.

- **Robust Multi-Target Domain Adaptation:** HDRQ consistently improves merging outcomes across different adaptation settings, confirming its effectiveness in real-world scenarios.

By addressing the impact of quantization through theoretical analysis and targeted regularization, HDRQ ensures that quantized models remain merge-compatible, paving the way for real-time adaptive AI on resource-constrained devices.

## 2. Related Work

### 2.1. Quantization

Quantization is a crucial technique for reducing model size and computational cost, making deep learning models more practical for resource-constrained environments. Research on quantization can be broadly categorized into two approaches: Quantization-Aware Training (QAT) and Post-Training Quantization (PTQ).

The first category, **Quantization-Aware Training (QAT)** (Esser et al., 2020; Baskin et al., 2021; Défossez et al., 2022; Shin et al., 2023), fine-tunes full-precision models using fake quantization operators and straight-through estimators (STE) (Bengio et al., 2013) to approximate gradients. These methods leverage the full dataset to compensate for errors induced by quantization. A notable subfield within QAT is noise-based quantization (Baskin et al., 2021; Défossez et al., 2022; Shin et al., 2023), which models quantization noise as an additive perturbation to weights, eliminating the need for STE and improving stability.

The second category, **Post-Training Quantization (PTQ)**, enables quantization without full retraining, making it more efficient but often at the cost of performance degradation. Various techniques have been developed to enhance PTQ: (Nagel et al., 2020) proposed a theoretically justified layer-wise reconstruction method to optimize rounding policies, while (Li et al., 2021) extended this idea to block-wise reconstruction, addressing cross-layer dependencies. (Wei et al., 2022) further introduced selective activation quantization and dropout strategies to enhance robustness. A more recent advancement, Bit-shrinking (Lin et al., 2023), leveraged noise-based quantization with sharpness-aware scheduling to minimize degradation. Our proposed method, **HDRQ**, falls within this category, introducing a compact noise-based quantization strategy specifically designed to maintain merging compatibility across models.

### 2.2. Domain Adaptation

Domain adaptation enables models to generalize to a target domain by leveraging knowledge from a related source domain. One of the most widely studied areas is **Unsupervised Domain Adaptation (UDA)** (Long et al., 2016; Zou et al., 2018), which aligns domain distributions using labeled source data and unlabeled target data. However, conventional UDA techniques become impractical when source data is unavailable due to privacy constraints. To address this, **Source-Free Domain Adaptation (SFDA)** (Liang et al., 2020; Liu et al., 2021; Hou & Zheng, 2021) has emerged, where adaptation relies solely on a pre-trained model and target domain data, eliminating the need for direct source data access.

Most domain adaptation approaches focus on **Single-Target Domain Adaptation (STDA)**, where a model adapts to one specific target domain. However, real-world applications often require adaptation to multiple distinct target domains. **Multi-Target Domain Adaptation (MTDA)** (Yu et al.,

2018; Gholami et al., 2020; Nguyen-Meidine et al., 2021; Li et al., 2024) addresses this by training a model capable of handling diverse target domains. Many MTDA methods employ multiple student models (Nguyen-Meidine et al., 2021), which significantly increases computational overhead.

A recent alternative, **training-free MTDA via model merging** (Li et al., 2024), leverages the observation that models fine-tuned from the same initialization often reside within a similar optimization basin. This enables weight merging via simple averaging, provided that normalization statistics are properly handled. While effective, this approach has largely overlooked the impact of quantization, which disrupts weight alignment and hinders merging quality.

Our work bridges this gap by proposing **HDRQ**, a quantization method explicitly designed to preserve merging compatibility in multi-target domain adaptation. By addressing quantization-induced misalignment, HDRQ unlocks new possibilities for real-time, adaptive AI on resource-constrained devices.

## 3. Analysis

Both quantization and model merging have been extensively explored, yet a rigorous understanding of how quantization noise affects model merging remains unexplored. To address this gap, we provide a theoretical analysis of quantization-induced misalignment and its impact on the merging process. Inspired by prior works (Ainsworth et al., 2023; Xu et al., 2024; Stoica et al., 2024) on model merging, we extend the concept of the error barrier to explicitly incorporate quantization effects. While previous studies have examined merging under full-precision settings, we analyze how quantization-induced perturbations affect the loss landscape and merging quality. This analysis reveals key factors that degrade merging performance and motivates our proposed quantization regularization techniques.

### 3.1. General error barrier case

An ideal merging process should ensure that interpolated models maintain low error without introducing sharp increases in loss. This motivates us to analyze the merging process through the lens of the *error barrier*, which quantifies the degree of interpolation-induced performance degradation. Given two converged weight points $\theta_1$ and $\theta_2$, we define the interpolated model as:

$$\theta_\lambda = (1 - \lambda)\theta_1 + \lambda\theta_2, \quad \lambda \in [0, 1]. \quad (1)$$

The error barrier (Frankle et al., 2020) is then given by:

$$\max_{\lambda \in [0,1]}[\mathcal{L}(\theta_\lambda) - \frac{1}{2}(\mathcal{L}(\theta_1) + \mathcal{L}(\theta_2))]. \quad (2)$$

The error barrier quantifies the maximum increase in error relative to the average loss along the linear path connecting the two points. It serves as an indicator of convexity (Frankle et al., 2020; Ainsworth et al., 2023). Specifically, a zero error barrier implies linear mode connectivity, indicating that the loss remains flat or exhibits positive curvature along the path. In other words, it provides insight into merging feasibility. A low error barrier indicates a smooth loss landscape, while a high barrier suggests weight misalignment.

Since the error induced by quantization can be approximated as the addition of uniform noise to the original values (Baskin et al., 2021; Défossez et al., 2022; Shin et al., 2023), we can derive the error barrier for the quantized weights as follows:

$$\max_{\lambda \in [0,1]}[\mathcal{L}(\theta_\lambda + \epsilon_\lambda) - \frac{1}{2}(\mathcal{L}(\theta_1 + \epsilon_1) + \mathcal{L}(\theta_2 + \epsilon_2))], \quad (3)$$

where $\epsilon_1, \epsilon_2$ represent quantization noise sampled from a uniform distribution $\epsilon_1 \sim U[-\frac{s_1}{2}, \frac{s_1}{2}]$ and $\epsilon_2 \sim U[-\frac{s_2}{2}, \frac{s_2}{2}]$, respectively, with quantization step sizes $s_1$ and $s_2$.

Applying a second-order Taylor expansion, we obtain:

$$\max_{\lambda \in [0,1]}[\mathcal{L}(\theta_\lambda) - \frac{1}{2}(\mathcal{L}(\theta_1) + \mathcal{L}(\theta_2)] +$$
$$\max_{\lambda \in [0,1]}[\epsilon_\lambda \cdot \nabla_\theta \mathcal{L}(\theta_\lambda) + \frac{1}{2}\epsilon_\lambda^T \cdot \nabla_\theta^2 \mathcal{L}(\theta_\lambda) \cdot \epsilon_\lambda -$$
$$\frac{1}{2}(\epsilon_1 \cdot \nabla_\theta \mathcal{L}(\theta_1) + \frac{1}{2}\epsilon_1^T \cdot \nabla_\theta^2 \mathcal{L}(\theta_1) \cdot \epsilon_1 +$$
$$\epsilon_2 \cdot \nabla_\theta \mathcal{L}(\theta_2) + \frac{1}{2}\epsilon_2^T \cdot \nabla_\theta^2 \mathcal{L}(\theta_2) \cdot \epsilon_2)]. \quad (4)$$

This yields the sum of the original error barrier and the maximum of terms involving the first- and second-order derivatives at the two points and their merged point. Given that both $\theta_1$ and $\theta_2$ have converged to the same loss $\mathcal{L}$, all terms involving the first-order derivatives can be ignored. Furthermore, assuming a zero loss barrier for the original points for simplicity, we obtain:

$$\max_{\lambda \in [0,1]}[\epsilon_\lambda \cdot \nabla_\theta \mathcal{L}(\theta_\lambda) + \frac{1}{2}\epsilon_\lambda^T \cdot \nabla_\theta^2 \mathcal{L}(\theta_\lambda) \cdot \epsilon_\lambda -$$
$$\frac{1}{4}(\epsilon_1^T \cdot \nabla_\theta^2 \mathcal{L}(\theta_1) \cdot \epsilon_1 + \epsilon_2^T \cdot \nabla_\theta^2 \mathcal{L}(\theta_2) \cdot \epsilon_2)]. \quad (5)$$

To minimize this term, we consider two approaches. The first approach maximizes the sum of the second-order terms. Since both weights are at local minima, their Hessians are positive semi-definite, ensuring that these terms remain non-negative regardless of $\epsilon_1$ and $\epsilon_2$. However, maximizing this term is undesirable, as an increased Hessian implies reduced robustness. This approach deliberately increases the loss of the quantized models to reduce the maximum difference between their mean and the interpolated point.

An alternative, but promising approach is to minimize the term related to the merged point $\theta_\lambda$. Assuming that the

Hessian of the loss $\mathcal{L}$ is $M$-Lipschitz continuous between $\theta_1$ and $\theta_2$, we can bound the Hessian at the merged point using the original points as follows:

$$|\nabla_\theta^2 \mathcal{L}(\theta_\lambda) - \frac{\nabla_\theta^2 \mathcal{L}(\theta_1) + \nabla_\theta^2 \mathcal{L}(\theta_2)}{2}| \leq \frac{M||\theta_2 - \theta_1||}{2}. \quad (6)$$

This result indicates that the Hessian at the merged point can be effectively regularized by controlling the Hessians at the original points and minimizing the distance between them. Since the Hessian is $M$-Lipschitz continuous, the gradient of the loss also becomes Lipschitz continuous with some constant $L$. Given that the Hessian is closely related to the rate of change of the gradient, regularizing both the Hessians and the distance between the points implicitly regularizes the first-order terms at the two points and the merged point.

## 3.2. Regularization for Merge-Friendly Quantization

Our theoretical analysis identifies two key contributors to the error barrier: (1) increased sensitivity to quantization noise due to sharp loss landscapes, and (2) excessive divergence between quantized weights that disrupt interpolation. Based on these insights, we introduce two regularization:

- **Hessian Regularization:** To reduce sensitivity to perturbations, we minimize the second-order term in (7) by encouraging smooth Hessian spectra during quantization. This prevents excessive local curvature that amplifies quantization noise effects.

- **Distance Regularization:** To control weight divergence, we minimize $||\theta_1 - \theta_2||$ during quantization, ensuring better alignment between the merged models.

By integrating these regularization techniques, we significantly mitigate the impact of quantization on merging performance. Figure 2 illustrates the effectiveness of HDRQ in flattening the loss landscape, leading to improved merging.

## 3.3. Domain Adaptation case

In domain adaptation, the losses are not necessarily equal, i.e., $\mathcal{L}(\theta_1) \not\approx \mathcal{L}(\theta_2)$, as the models are optimized with respect to different domain-specific objectives. This discrepancy shifts the lower bound of the error barrier from 0 to $\frac{1}{2}|(\mathcal{L}(\theta_1) - \mathcal{L}(\theta_2))|$. Despite this shift, minimizing the error barrier remains essential for effective model merging.

The key implication of this change is that one of the first-order terms at the original points in Equation (7) does not vanish. However, by leveraging the fact that domain adaptation from the same source weight results in weights located within a single basin (Li et al., 2024), the remaining first-order term can be absorbed into that of the merged point. Let us assume $\theta_1$ and $\theta_2$ are obtained through domain adaptation from the same source weight $\theta_0$, optimized with losses

$L_1$ and $L_2$, respectively. For simplicity, we analyze the case from the perspective of one domain with loss $L_1$, though the same reasoning applies symmetrically to the other domain.

When the Hessian is $M$-Lipschitz continuous, as assumed in the general error barrier case, the gradient also becomes Lipschitz continuous with some constant $L$. Since $\theta_1$ and $\theta_2$ lie within the same basin, their linear interpolation $\theta_\lambda$ also resides within this basin. Consequently, the Jacobian term $\nabla_\theta L_1(\theta_2)$ in Equation (7) becomes a scaled version of $\nabla_\theta \theta_\lambda$, proportional to the distance between points. Therefore, Equation (7) can be reformulated as:

$$\max_{\lambda \in [0,1]} [L_1(\theta_\lambda) - \frac{1}{2}(\mathcal{L}_1(\theta_1) + \mathcal{L}_1(\theta_2)] +$$

$$\max_{\lambda \in [0,1]} [(\epsilon_\lambda + k \cdot \epsilon_2) \cdot \nabla_\theta \mathcal{L}_1(\theta_\lambda) + \frac{1}{2} \epsilon_\lambda^T \cdot \nabla_\theta^2 \mathcal{L}_1(\theta_\lambda) \cdot \epsilon_\lambda -$$

$$\frac{1}{4}(\epsilon_1^T \cdot \nabla_\theta^2 \mathcal{L}_1(\theta_1) \cdot \epsilon_1 + \epsilon_2^T \cdot \nabla_\theta^2 \mathcal{L}_1(\theta_2) \cdot \epsilon_2)], \quad (7)$$

where $k$ is a scalar proportional to the distance between $\theta\lambda$ and $\theta_2$. Although it may not be feasible for each weight to directly account for the Hessians of both losses, regularizing it within a single domain can still indirectly regulate the upper bound of the error at the merged points. This insight ensures that the model remains more robust to quantization-induced errors and facilitates smoother merging across domains.

# 4. HDRQ: Hessian and Distance Regularizing Quantization

Building upon our previous analysis, we propose a novel quantization method called Hessian and Distance Regularizing Quantization (HDRQ). Our method incorporates two key strategies to enable merge-friendly quantized networks. First, it employs noise-based quantization to regularize the Hessian, reducing the sensitivity of the loss landscape to weight perturbations. Second, it introduces weight distance regularization, encouraging quantized models to converge that are inherently more compatible for merging.

Additionally, to address the rounding ambiguity problem that occurs during the merging of quantized models, we propose an advanced noise sampling-based rounding technique. This approach effectively mitigates ambiguity in rounding policies, ensuring stable weight merging and enhancing the overall quality of the adapted models.

## 4.1. Noise-based hessian regularization

To regularize the Hessian of the network, HDRQ simulates quantization by introducing additive sampled quantization noise. For each optimization step involving weight $w$ and quantization step size $\Delta$, we first compute the quantized weight $\hat{w}$ under uniform quantization as follows:

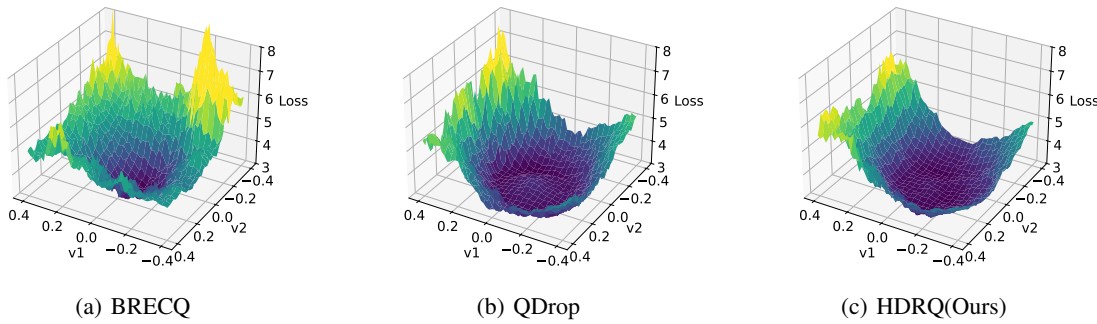

(a) BRECQ        (b) QDrop        (c) HDRQ(Ours)

*Figure 2.* Visualization of loss surfaces quantized with each method is shown. ResNet-50 adapted from Real domain to Clipart domain (R → C) is quantized to W4A8. HDRQ effectively regularize hessian with noise-based quantization, leading weights to flatter surface.

$$\hat{w} = clamp(\lfloor \frac{w}{\Delta} \rceil, -2^{b-1}, 2^{b-1} - 1) \cdot \Delta, \qquad (8)$$

where $b$ denotes the bit width. We then sample the quantization noise $\epsilon$ from the quantization error $w - \hat{w}$ and train using $w + \epsilon$ instead of the deterministic quantized value $\hat{w}$. Since the quantization noise follows a uniform distribution $U[-\frac{\Delta}{2}, \frac{\Delta}{2}]$, the modified loss function inherently regularizes the Hessian as follows:

$$E[\mathcal{L}(\hat{w})] \approx E[\mathcal{L}(w + e)]$$
$$\approx E[\mathcal{L}(w) + \epsilon \cdot \nabla_w \mathcal{L}(w) + \frac{1}{2}\epsilon^T \cdot \nabla_w^2 \mathcal{L}(w) \cdot \epsilon]$$
$$\approx E[\mathcal{L}(w) + \frac{1}{2}\epsilon^T \cdot \nabla_w^2 \mathcal{L}(w) \cdot \epsilon], \qquad (9)$$

where the first-order term of the Taylor expansion vanishes because $E[\epsilon] = 0$. As a result, the expected loss function penalizes sharp curvature in the loss landscape, leading to implicit Hessian regularization.

While noise-based quantization is not a novel concept—its effectiveness has been established in prior works (Baskin et al., 2021; Défossez et al., 2022; Shin et al., 2023; Lin et al., 2023)—our approach is the first to integrate this technique within a model merging framework, motivated by theoretical analysis. This integration results in smoother loss surfaces and improved compatibility between quantized models during merging.

The effect of noise-based Hessian regularization is demonstrated in Figure 2. When this regularization technique is applied, the network converges to a smoother loss surface compared to existing methods. This smoother convergence not only enhances the robustness of the quantized model but also facilitates better merging compatibility by reducing sharp loss barriers between models.

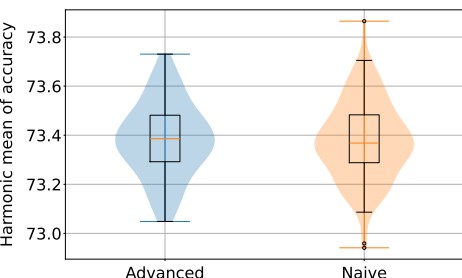

*Figure 3.* The distribution of harmonic mean accuracy for merging W4A8 quantized C→ R and C→ A models on the Office-Home dataset is presented. Our simple yet effective cosine similarity-based method, denoted as Advanced, successfully filters out low-quality weights, stabilizing merging outcomes.

### 4.2. Weight distance regularization

Measuring the distance between separately adapted weights for each domain is challenging, as no prior information about the target domain is assumed. Instead of directly minimizing this distance, HDRQ ensures that the distances between domain-adapted weights remain small by regularizing their distances from the source weight. The upper bound of the distance between the two domain-adapted weights, $w_{tar1}$ and $w_{tar2}$, can be derived using the triangular inequality:

$$|w_{tar1} - w_{tar2}| \leq |w_{src} - w_{tar1}| + |w_{src} - w_{tar2}|. \quad (10)$$

Since both the source and domain-adapted weights reside within the same basin, enforcing this regularization during quantization is unlikely to significantly degrade weight quality. To implement this, we introduce a regularization term based on the $l_2$-norm between the source and target weights.

It is important to note that assuming access to the original weights is reasonable. In most deployment scenarios, models are initially pretrained on source data and subsequently adapted on user devices using user-specific data.

*Table 1.* Quantization and multi-target domain adaptation results on semantic segmentation task

| Method | Bit(W/A) | Domain | road | sidewalk | building | wall | fence | pole | traffic light | traffic sign | vegetation | terrain | sky | person | rider | car | truck | bus | train | motorcycle | bicycle | mIoU |
|---|---|---|---|---|---|---|---|---|---|---|---|---|---|---|---|---|---|---|---|---|---|---|
| FP | 32/32 | $G \to C$ | 95.71 | 72.37 | 88.52 | 27.88 | 22.65 | 49.62 | 59.37 | 69.97 | 89.47 | 44.69 | 89.63 | 77.37 | 49.26 | 91.29 | 54.30 | 56.65 | 38.22 | 36.91 | 58.25 | 61.69 |
| | | $G \to I$ | 97.23 | 4.41 | 80.91 | 38.86 | 11.03 | 34.70 | 21.50 | 44.18 | 80.83 | 39.80 | 95.71 | 72.56 | 63.62 | 80.25 | 58.56 | 60.16 | 0.00 | 78.63 | 26.13 | 52.06 |
| BRECQ | 6/6 | $G \to C$ | 95.85 | 71.73 | 88.44 | 25.12 | 23.72 | 48.83 | 57.46 | 69.42 | 89.24 | 43.63 | 89.39 | 76.78 | 47.82 | 91.40 | 54.53 | 55.48 | 36.65 | 33.76 | 57.17 | 60.86 |
| | | $G \to I$ | 97.15 | 5.40 | 80.55 | 38.57 | 13.90 | 35.48 | 16.98 | 43.58 | 80.51 | 39.09 | 95.51 | 72.36 | 63.35 | 80.09 | 58.52 | 59.23 | 0.00 | 77.97 | 25.92 | **51.80** |
| QDrop | 6/6 | $G \to C$ | 95.54 | 71.41 | 88.49 | 28.33 | 23.82 | 48.87 | 58.38 | 70.06 | 89.52 | 44.63 | 89.73 | 76.79 | 48.04 | 91.19 | 53.13 | 57.66 | 35.94 | 35.89 | 57.62 | 61.32 |
| | | $G \to I$ | 97.13 | 4.61 | 80.70 | 39.01 | 11.60 | 34.05 | 17.87 | 44.21 | 80.63 | 39.69 | 95.62 | 71.99 | 63.37 | 79.95 | 59.01 | 59.96 | 0.00 | 78.18 | 23.82 | 51.65 |
| Ours | 6/6 | $G \to C$ | 95.64 | 71.83 | 88.47 | 29.46 | 23.11 | 49.33 | 57.98 | 69.96 | 89.45 | 44.22 | 89.57 | 76.72 | 47.32 | 91.32 | 55.12 | 56.35 | 35.60 | 39.23 | 58.61 | **61.54** |
| | | $G \to I$ | 97.20 | 7.34 | 80.60 | 38.80 | 13.39 | 34.68 | 13.55 | 43.53 | 80.75 | 39.97 | 95.65 | 71.34 | 62.69 | 79.68 | 58.95 | 59.61 | 0.00 | 78.00 | 23.41 | 51.53 |
| BRECQ | 4/4 | $G \to C$ | 90.36 | 41.90 | 83.73 | 12.16 | 24.78 | 42.92 | 43.57 | 57.77 | 87.96 | 40.64 | 88.05 | 72.89 | 40.63 | 81.28 | 45.13 | 48.32 | 36.81 | 28.63 | 52.27 | 53.67 |
| | | $G \to I$ | 94.90 | 2.01 | 71.59 | 27.91 | 19.49 | 28.61 | 2.53 | 35.70 | 77.04 | 34.97 | 91.49 | 60.66 | 58.42 | 78.41 | 55.01 | 47.64 | 0.00 | 68.97 | 9.05 | 45.50 |
| QDrop | 4/4 | $G \to C$ | 94.94 | 69.01 | 87.85 | 24.99 | 24.55 | 46.99 | 53.66 | 68.27 | 88.86 | 43.07 | 88.94 | 74.53 | 43.87 | 89.95 | 49.29 | 54.44 | 27.51 | 32.76 | 56.04 | **58.92** |
| | | $G \to I$ | 96.68 | 2.48 | 79.38 | 36.29 | 12.31 | 32.32 | 10.40 | 42.19 | 80.39 | 39.97 | 95.22 | 69.19 | 61.52 | 78.48 | 56.80 | 54.87 | 0.00 | 75.98 | 14.90 | **49.44** |
| Ours | 4/4 | $G \to C$ | 95.23 | 69.78 | 87.63 | 26.60 | 20.82 | 46.93 | 51.65 | 67.90 | 89.04 | 42.25 | 89.12 | 73.63 | 38.86 | 90.13 | 47.50 | 52.09 | 25.88 | 34.97 | 56.46 | 58.23 |
| | | $G \to I$ | 96.74 | 3.52 | 79.03 | 35.56 | 11.54 | 33.04 | 7.22 | 40.18 | 79.71 | 38.76 | 95.41 | 66.29 | 59.74 | 78.23 | 56.89 | 55.09 | 0.00 | 75.67 | 12.29 | 48.68 |

(a) Quantization results on each target domain

| Method | Bit(W/A) | Metric | road | sidewalk | building | wall | fence | pole | traffic light | traffic sign | vegetation | terrain | sky | person | rider | car | truck | bus | train | motorcycle | bicycle | mIoU |
|---|---|---|---|---|---|---|---|---|---|---|---|---|---|---|---|---|---|---|---|---|---|---|
| FP | 32/32 | C | 91.68 | 49.98 | 86.83 | 36.25 | 38.83 | 46.82 | 54.05 | 55.10 | 89.31 | 42.77 | 90.18 | 75.37 | 41.86 | 89.99 | 50.51 | 58.04 | 22.46 | 35.38 | 48.86 | 58.12 |
| | | I | 97.21 | 33.64 | 80.12 | 35.85 | 24.95 | 32.95 | 24.76 | 41.86 | 79.45 | 34.70 | 94.54 | 73.88 | 60.97 | 78.01 | 51.37 | 57.71 | 0.00 | 76.96 | 37.65 | 53.50 |
| | | H | 94.36 | 40.21 | 83.34 | 36.05 | 30.38 | 38.68 | 33.96 | 47.58 | 84.09 | 38.31 | 92.31 | 74.62 | 49.64 | 83.57 | 50.94 | 57.87 | 0.00 | 48.48 | 42.53 | 55.71 |
| BRECQ | 6/6 | C | 87.51 | 37.85 | 84.99 | 23.90 | 31.28 | 42.44 | 48.39 | 49.52 | 87.78 | 36.48 | 87.56 | 73.02 | 39.78 | 78.03 | 45.79 | 53.45 | 16.66 | 29.20 | 44.35 | 52.52 |
| | | I | 96.90 | 32.24 | 78.21 | 29.63 | 21.66 | 31.31 | 22.24 | 38.83 | 78.86 | 35.78 | 93.76 | 71.12 | 58.71 | 74.51 | 45.03 | 53.68 | 0.00 | 75.09 | 33.85 | 51.12 |
| | | H | 91.94 | 34.55 | 81.45 | 26.36 | 25.51 | 36.01 | 30.38 | 43.36 | 83.07 | 35.78 | 90.54 | 72.05 | 47.38 | 76.04 | 45.31 | 53.52 | 0.00 | 41.74 | 38.37 | 51.80 |
| QDrop | 6/6 | C | 87.44 | 40.12 | 85.08 | 26.04 | 31.94 | 41.82 | 48.95 | 52.07 | 88.20 | 37.60 | 88.60 | 73.27 | 38.60 | 76.99 | 47.67 | 55.27 | 19.81 | 28.42 | 43.40 | 53.23 |
| | | I | 96.72 | 32.46 | 78.58 | 30.58 | 23.90 | 31.57 | 26.10 | 41.38 | 78.99 | 33.11 | 94.16 | 72.06 | 59.50 | 74.41 | 46.93 | 54.44 | 0.00 | 75.09 | 34.64 | 51.82 |
| | | H | 91.83 | 35.77 | 81.69 | 28.04 | 27.25 | 35.95 | 33.93 | 46.10 | 83.34 | 34.84 | 91.29 | 72.66 | 46.81 | 75.50 | 47.25 | 54.84 | 0.00 | 41.14 | 38.51 | 52.51 |
| Ours | 6/6 | C | 89.38 | 47.97 | 85.19 | 29.36 | 34.05 | 43.76 | 50.22 | 54.24 | 88.57 | 37.92 | 88.75 | 73.62 | 37.92 | 82.34 | 48.63 | 55.45 | 19.29 | 30.00 | 44.58 | 54.88 |
| | | I | 97.48 | 33.35 | 78.82 | 32.82 | 23.84 | 31.86 | 25.15 | 42.09 | 78.83 | 36.55 | 94.39 | 72.42 | 59.65 | 76.45 | 48.35 | 54.81 | 0.00 | 76.06 | 36.18 | 52.58 |
| | | H | 93.25 | 39.25 | 81.87 | 30.95 | 28.02 | 36.86 | 33.45 | 47.38 | 83.41 | 37.84 | 91.48 | 73.01 | 46.34 | 79.24 | 48.45 | 55.11 | 0.00 | 42.86 | 39.94 | **53.70** |
| BRECQ | 4/4 | C | 60.65 | 5.48 | 67.05 | 4.87 | 8.51 | 13.56 | 22.22 | 17.46 | 70.92 | 13.59 | 71.41 | 43.65 | 8.59 | 59.05 | 17.26 | 28.61 | 8.14 | 16.01 | 17.93 | 29.21 |
| | | I | 84.71 | 18.07 | 56.39 | 12.74 | 13.03 | 18.91 | 8.46 | 23.55 | 70.04 | 24.72 | 74.94 | 48.00 | 38.98 | 54.66 | 27.67 | 25.98 | 0.00 | 53.86 | 16.80 | 35.34 |
| | | H | 70.06 | 8.09 | 60.55 | 6.90 | 9.90 | 15.58 | 12.08 | 19.37 | 70.40 | 16.70 | 72.51 | 45.53 | 13.74 | 56.43 | 20.32 | 26.76 | 0.00 | 24.25 | 16.98 | 31.95 |
| QDrop | 4/4 | C | 80.75 | 10.68 | 77.57 | 12.62 | 19.34 | 25.28 | 29.52 | 37.63 | 80.47 | 21.73 | 78.26 | 61.92 | 21.18 | 74.64 | 34.81 | 41.43 | 6.45 | 18.77 | 25.25 | 39.91 |
| | | I | 91.54 | 17.59 | 67.78 | 22.80 | 18.99 | 23.79 | 11.66 | 36.53 | 75.05 | 27.07 | 86.68 | 62.28 | 49.08 | 68.91 | 34.67 | 41.01 | 0.00 | 65.66 | 22.87 | 43.37 |
| | | H | 85.70 | 12.45 | 72.13 | 16.08 | 18.85 | 24.41 | 16.50 | 37.00 | 77.64 | 23.54 | 82.07 | 62.08 | 29.25 | 71.32 | 34.57 | 41.05 | 0.00 | 28.81 | 23.89 | 41.54 |
| Ours | 4/4 | C | 83.36 | 18.63 | 80.15 | 16.81 | 26.99 | 34.00 | 35.01 | 43.11 | 84.96 | 28.75 | 81.18 | 67.49 | 25.84 | 77.21 | 35.84 | 42.72 | 6.99 | 20.91 | 30.04 | 44.44 |
| | | I | 94.78 | 25.67 | 73.27 | 25.85 | 19.49 | 28.28 | 17.94 | 38.15 | 78.35 | 30.46 | 91.85 | 67.13 | 53.48 | 70.81 | 38.92 | 45.66 | 0.00 | 69.90 | 26.23 | 47.17 |
| | | H | 88.67 | 21.08 | 76.49 | 20.28 | 22.50 | 30.82 | 23.60 | 40.45 | 81.51 | 29.37 | 86.12 | 67.29 | 34.65 | 73.69 | 39.50 | 44.00 | 0.00 | 31.95 | 27.95 | **45.75** |

(b) Multi-target domain adaptation via model merging results. C and I represents CityScapes and Indian Driving Dataset, respectively. H denotes the harmonic mean of performance on C and I. HDRQ demonstrates comparable or superior performance across all settings.

## 4.3. Handling ambiguity in rounding policy

While HDRQ optimizes the network to a more merge-friendly state, naive merging may still lead to quality degradation due to ambiguity in the rounding policy.

Consider two quantized values being merged with integer representations $I_1$ and $I_2$, and step sizes $\Delta_1$ and $\Delta_2$, respectively. Using the midpoint merging technique described in (Li et al., 2024), if the sum of $I_1$ and $I_2$ is an odd number, the merged integer value falls between two adjacent integers, leading to ambiguity in the rounding direction. A potential solution is to merge in the floating-point domain as follows:

$$I_{merged} = \lfloor \frac{I_1 \cdot \Delta_1 + I_2 \cdot \Delta_2}{\Delta_1 + \Delta_2} \rceil. \qquad (11)$$

However, when the step sizes $\Delta_1$ and $\Delta_2$ are similar, this

approach again degenerates into an ambiguous case, as the $\Delta$ terms effectively cancel out. This issue is particularly pronounced in domain adaptation scenarios, where weights are fine-tuned from shared source weights with a small learning rate, making their step sizes likely to be similar.

To handle the ambiguity problem in rounding policies, HDRQ employs noise sampling during the merging process. Instead of merging weights directly in the integer or floating-point domain, we sample noise $\epsilon \sim U[-\frac{\Delta}{2}, \frac{\Delta}{2}]$ and add it to the weights prior to merging. This maintains the same quantized representation as the original values while mitigating ambiguity:

$$I_{merged} = \lfloor \frac{(I_1 \cdot \Delta_1 + \epsilon_1) + (I_2 \cdot \Delta_2 + \epsilon_2)}{\Delta_1 + \Delta_2} \rceil. \qquad (12)$$

*Table 2.* Multi-target domain adaptation results on the Office-Home dataset. The harmonic mean of accuracy across three target domains for the merged model is reported as the main metric, with individual domain performance of each quantized models provided in parentheses. The best accuracy for each configuration is highlighted in bold, with red text indicating a gap of more than $1\%$ compared to the second-best result. Blue text highlights cases where previous methods failed to converge to a flat loss surface. Note that sampling is not applied in this setting, as there is no ambiguity in merging across the three-task scenario.

| Domain | FP | Methods | W8A8 | W8A4 | W4A8 | W4A4 | W3A3 |
|---|---|---|---|---|---|---|---|
| R → A,C,P | 67.78 (73.59\|59.22\|83.87) | BRECQ | 67.74 (73.55\|59.18\|83.85) | 64.79 (73.09\|55.62\|82.92) | 64.15 (73.38\|59.11\|83.87) | 60.95 (72.89\|56.56\|82.70) | 43.66 (67.00\|47.74\|73.49) |
| | | QDrop | **67.91** (73.51\|59.20\|83.83) | **67.4** (73.34\|59.24\|83.78) | 64.85 (73.71\|59.15\|83.92) | 66.26 (73.47\|58.99\|83.92) | 62.99 (73.14\|58.69\|83.62) |
| | | HDRQ | 67.46 (73.34\|59.06\|83.85) | 66.71 (73.34\|59.18\|83.71) | **66.74** (73.42\|59.20\|83.89) | **66.41** (73.42\|59.04\|83.87) | **64.70** (72.81\|58.72\|83.33) |
| A → R,C,P | 68.80 (81.78\|55.99\|78.06) | BRECQ | **68.75** (81.71\|56.01\|78.06) | 65.4 (80.06\|54.91\|77.16) | 66.06 (81.73\|55.99\|77.99) | 62.53 (80.35\|55.49\|77.11) | 48.04 (71.43\|48.84\|71.32) |
| | | QDrop | **68.75** (81.71\|55.97\|78.04) | **68.24** (81.75\|55.85\|78.15) | 66.83 (81.66\|55.88\|78.04) | 66.04 (81.80\|55.78\|78.13) | 64.22 (81.41\|55.67\|77.88) |
| | | HDRQ | 68.10 (81.73\|55.92\|77.97) | 68.15 (81.75\|55.78\|77.97) | **67.80** (81.68\|55.92\|78.04) | **67.58** (81.89\|55.83\|77.83) | **65.29** (80.97\|55.40\|78.04) |
| C → R,A,P | 75.07 (79.11\|68.97\|79.43) | BRECQ | 74.75 (79.07\|68.89\|79.39) | 73.51 (78.70\|68.81\|79.07) | 73.22 (79.11\|69.06\|79.39) | 71.31 (78.49\|68.81\|78.91) | 56.16 (72.69\|64.69\|70.40) |
| | | QDrop | **74.79** (79.09\|68.89\|79.43) | **74.58** (79.23\|68.85\|79.45) | 73.81 (79.11\|68.89\|79.36) | 73.25 (79.23\|68.69\|79.39) | 71.01 (79.07\|68.40\|79.25) |
| | | HDRQ | 74.58 (79.14\|69.14\|79.34) | 73.70 (79.21\|68.69\|79.14) | **74.26** (79.23\|68.77\|79.39) | **73.58** (79.11\|68.60\|79.18) | **71.63** (78.77\|68.07\|78.96) |
| P → R,A,C | 65.25 (82.03\|68.27\|55.95) | BRECQ | **65.14** (81.96\|68.19\|55.90) | 63.27 (80.88\|67.94\|54.73) | **64.09** (82.03\|68.15\|56.06) | 61.92 (81.25\|67.82\|53.97) | 45.09 (72.11\|62.75\|41.90) |
| | | QDrop | 64.8 (81.98\|68.27\|55.97) | **64.52** (82.05\|67.86\|55.99) | 62.52 (82.10\|68.19\|55.88) | **63.22** (81.89\|68.07\|56.08) | 61.24 (81.68\|67.82\|55.51) |
| | | HDRQ | 64.51 (81.91\|68.11\|56.08) | 64.09 (81.96\|68.15\|56.01) | 63.93 (82.01\|68.19\|55.83) | 63.19 (82.10\|68.19\|55.90) | **61.55** (81.23\|67.45\|55.33) |

Since noise sampling introduces randomness, it is crucial to reject noise samples that could degrade the quality of the merged network. To achieve this, we evaluate the cosine similarity between the vector connecting the merged sampled weights to the target domain weights and the original interpolation vector connecting the target domain weights. The sample with the highest similarity is selected, ensuring high-quality merging. As shown in Figure 3, this straightforward yet effective approach successfully filters out low-quality weight samples and stabilizes the merging quality.

## 5. Experimental Results

To validate the effectiveness of our proposed HDRQ method, we conduct experiments on multi-target domain adaptation tasks for semantic segmentation and image classification. Following our initial objective, we adopt the merging-based multi-target domain adaptation approach (Li et al., 2024), where models are first adapted independently to each target domain and then merged for unified multi-target adaptation. Quantization is applied between the single-target domain adaptation and the merging process. The harmonic mean of performances across target domains is reported as the primary evaluation metric. Unless otherwise specified, advanced noise-based sampling is used during the merging step after quantization. We report the mean results from 30 sampled weights for semantic segmentation.

**Semantic Segmentation** For semantic segmentation, we use the GTA synthetic dataset (Richter et al., 2016) as the source domain and two real-world datasets, Cityscapes (Cordts et al., 2016) and Indian Driving datasets (Varma et al., 2019), as the target domains. We adopt HRDA (Hoyer et al., 2022) as the single-target domain adaptation method, using ResNet-101 (He et al., 2016) as the backbone model and simple convolution head. All other settings are kept consistent with the original paper.

**Image Classification** For image classification, we use the Office-Home dataset (Venkateswara et al., 2017), which consists of four domains (Real, Art, Clipart, and Product). We select one of the four domains as the source domain and treat the other three as target domains. ResNet-50 (He et al., 2016) is used as the backbone architecture, and SHOT (Liang et al., 2020) is adopted as the single-target domain adaptation method. Adaptation settings follow the original work.

**Quantization Details** Quantization is applied after folding batch normalization layers. Thus, batch normalization layers do not need to be considered and only mid-point weight averaging is used for the merging process. The hyperparameter $\lambda$ for weight distance regularization is set to 5e-2 for all tasks. HDRQ conducts 20,000 iterations of block-wise reconstruction, including activation quantization with partial dropout. We use the Adam optimizer with an initial learning rate of 0.001 and a cosine annealing with warmup. For the last 3,500 iterations of reconstruction, we switch from noise-based quantization simulation to actual fake quantization, which stabilizes training and improves the final results by decreasing the gap between simulated and actual quantization. It is important to note that most reconstruction is conducted with noise-based quantization, and the learning rate becomes very small at the final stage, ensuring this switch does not undermine our objectives.

### 5.1. Semantic Segmentation

The results for multi-target domain adaptation on semantic segmentation are presented in Section 4.1. HDRQ demonstrates comparable performance to QDrop for the quantization task itself, while the merged quantized models significantly outperform other methods after multi-target domain adaptation. The benefits of HDRQ become more pronounced as the bit precision decreases, where the impact of quantization on model merging becomes more severe. While HDRQ surpasses QDrop by a modest 1.19 mIoU in the W6A6 setting, it achieves a much more substantial improvement of 4.21 mIoU in the W4A4 setting.

These results highlight the importance of robust quantization strategies for successful multi-target domain adaptation via model merging and validate the effectiveness of the Hessian and distance regularizations proposed by HDRQ.

### 5.2. Office Home

The experimental results on the Office-Home dataset (Venkateswara et al., 2017) are summarized in Table 2. We report the harmonic mean of accuracies across target domains, with the best results highlighted in bold, and red-colored results indicating accuracy gains of over $> 1\%$ compared to the second-best result.

For the relatively easy-to-quantize Office-Home task, all methods achieve comparable performance when weight precision is high. However, in specific settings such as R $\rightarrow$ A,C,P, when weights are quantized into lower bit-width, previous methods exhibit significant performance degradation compared to HDRQ.

We conjecture that this discrepancy arises because previous methods occasionally fail to converge to flat minima, which are critical for successful merging after quantization. Since

*Table 3.* Incremental ablation study on each component of HDRQ

| Method | Accuracy |
|---|---|
| Baseline | 62.99 (73.14\|58.69\|83.62) |
| + Noise-based quantization | 64.21 (73.42\|58.65\|83.67) |
| + Distance regularization | 64.70 (72.81\|58.72\|83.33) |

block-wise reconstruction and partial dropping may regularize overall hessian of loss landscape, it may not effective to handle lumps around actual converged points. In contrast, HDRQ effectively guides the network to smoother loss surfaces by direct injection of noise to weights, as visualized in Figure 2. This ability to find flatter minima is particularly evident in the P $\rightarrow$ R, A, C task, where QDrop struggles to merge the models successfully. HDRQ, however, demonstrates robust and consistent performance, achieving either comparable or superior results across all evaluated settings.

### 5.3. Ablation Study

To evaluate the effectiveness of each component in HDRQ, we conducted an incremental ablation study on the Office-Home dataset under the W3A3 precision, R $\rightarrow$ A,C,P setting. The results are presented in Table 3.

When neither noise-based quantization nor weight distance regularization was applied, our method degenerated to QDrop, which only employs block-wise reconstruction and partial dropping during quantization. Introducing the noise-based quantization scheme yielded a significant performance gain of 1.22% , highlighting the importance of direct regularization around the converged solution surface.

Further incorporating weight distance regularization brought an additional 0.49% accuracy gain, showing the effectiveness of keeping target domain weights close to each other for improved merging stability. These results underscore the combined value of noise-based quantization and distance regularization in HDRQ for enhancing model merging.

## 6. Conclusion

In this paper, we analyze the impact of quantization on model merging, particularly its effect on the error barrier. Building on this, we propose HDRQ, a post-training quantization scheme tailored for multi-target domain adaptation. HDRQ leverages noise-based quantization to regularize the Hessian and applies weight distance regularization to facilitate better merging. It also mitigates the rounding ambiguity inherent in naive merging methods through noise sampling. Experimental results demonstrate that HDRQ consistently outperforms previous approaches, validating its superiority.

## Acknowledgements

This work was supported by IITP and NRF grant funded by the Korea government(MSIT) (No. RS-2019-II191906, RS-2024-00396013, RS-2023-00228970, RS-2024-00457882).

## Impact Statement

This paper addresses the novel problem of combining quantization with multi-target domain adaptation and proposes an effective solution. Our work holds potential societal contributions by advancing the practical deployment of machine learning models with reduced computational and memory footprints.

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
