# OpenReview forum: "Merge-Friendly Post-Training Quantization for Multi-Target Domain Adaptation"
_ICML.cc/2025/Conference — ICML 2025 poster_

### Official Review · Reviewer_zjyC · 2025-02-27

**Overall Recommendation:** 2

**Summary:**

This paper proposes a post-training quantization (PTQ) method tailored specifically for the model merging scenario in multi-target domain adaptation. In particular, the paper argues that existing PTQ methods quantize each individual model into a state that makes them difficult to merge effectively into a single multi-domain model. To address this issue, the proposed method, HDRQ (Hessian and Distance Regularizing Quantization), introduces three specific design elements aimed at mitigating the degradation in merging quality. These include novel regularization strategies based on Hessian and weight-distance measures, as well as an advanced noise-sampling-based rounding technique.

**Claims And Evidence:**

The authors claim that conventional PTQ methods lead to failure in model merging due to uncontrolled weight perturbations, and they propose a “merge-friendly” PTQ method supported by both theoretical analysis and empirical evidence. However, several issues arise:

- Limited Merging Setting:
Both the theoretical analysis and experiments consider only interpolated models—the most naïve strategy for model merging. In reality, many state-of-the-art (SOTA) merging methods exist and should be included for a more robust comparison. (See the “Essential References Not Discussed” section for relevant literature.) This limitation undermines the motivation for HDRQ, as it remains unclear whether the observed degradation is inherent to PTQ or specific to the merging strategy chosen.

- Attribution of Accuracy Degradation:
In the experimental section (e.g., Table 1(b) and Table 2), it is observed that merged quantized models yield slightly lower performance (e.g., in terms of mIoU) compared to merged full-precision models. However, it is not clear whether this drop is due to the quantization process itself or because the quantized weights are less amenable to merging. For instance, Table 1(a) shows that quantization alone degrades performance even without merging. Hence, the authors should clarify whether the accuracy drop after merging quantized models is a consequence of quantization per se or of the incompatibility of the quantized weights with the merging process.

- Baseline Performance:
More critically, the experimental results indicate that the proposed HDRQ method does not consistently outperform the compared baseline; in particular, Table 2 suggests that QDrop yields superior performance. This raises concerns regarding the superiority of HDRQ over existing methods.

**Essential References Not Discussed:**

Representative papers about model merging:
[1] Yadav, Prateek, et al. "Ties-merging: Resolving interference when merging models." NeurIPS 2023.
[2] Huang, Chenyu, et al. "Emr-merging: Tuning-free high-performance model merging." NeurIPS 2024.
[3] Ilharco, Gabriel, et al. "Editing models with task arithmetic." ICLR 2023.
[4] Wang, Ke, et al. "Localizing task information for improved model merging and compression." ICML 2024.


SOTA papers about PTQ:
[1] Lee, Jung Hyun, et al. "Flexround: Learnable rounding based on element-wise division for post-training quantization." ICML 2023.
[2] Shang, Yuzhang, et al. "Enhancing post-training quantization calibration through contrastive learning." CVPR 2024.
[3] Xu, Ke, et al. "Ptmq: Post-training multi-bit quantization of neural networks." AAAI 2024.

**Experimental Designs Or Analyses:**

As noted in the Methods and Evaluation Criteria section, the benchmarks employed in the paper require a more formal treatment. In addition, there are two major concerns regarding the experimental design:

- The PTQ methods used for comparison are not state-of-the-art. For instance, BRECQ (from 2021) and QDrop (from 2022) are used, yet these methods are relatively outdated—and they have not been properly cited in the main text. More recent approaches (see the Essential References Not Discussed section) should be considered for a fair comparison.

- The model merging strategy evaluated in the experiments is limited solely to model interpolation—a very naïve merging approach. There exist many advanced, state-of-the-art merging methods that should be included in the experiments to thoroughly validate the proposed approach.

Moreover, the reported experimental results are not statistically significant. As mentioned in the Claims and Evidence section, the results in Table 1(a) and Table 2 do not clearly demonstrate HDRQ’s superiority; in fact, the baseline QDrop appears to outperform HDRQ.

**Methods And Evaluation Criteria:**

The proposed HDRQ method is derived through theoretical analysis and is based on a well-reasoned process. However, as mentioned above, the theoretical derivation is based on a setting where model merging is confined to interpolated models—a scenario that is overly simplistic and not representative of more advanced merging cases. Moreover, the experimental evaluation does not incorporate SOTA merging methods.

Furthermore, since the paper’s goal is to develop a PTQ method that is “merge-friendly,” the benchmarks should be drawn from established model merging benchmarks. The authors are encouraged to evaluate HDRQ across a variety of vision tasks, NLP tasks, and different numbers of target tasks, as well as on models fine-tuned via parameter-efficient fine-tuning (PEFT). The authors may want to learn from the following papers:

[1] Yadav, Prateek, et al. "Ties-merging: Resolving interference when merging models." NeurIPS 2023.
[2] Huang, Chenyu, et al. "Emr-merging: Tuning-free high-performance model merging." NeurIPS 2024.

**Other Comments Or Suggestions:**

Please refer to the other sections. Thanks.

**Other Strengths And Weaknesses:**

Please refer to the other sections. Thanks.

**Questions For Authors:**

Please refer to the other sections. Thanks.

**Relation To Broader Scientific Literature:**

N/A

**Theoretical Claims:**

As mentioned in the Claims and Evidence section, the theoretical claims in this paper have a major issue: the basic setting is not entirely realistic because it only considers interpolated models. Apart from this initial setting, the subsequent derivations are straightforward and easily understandable. Consequently, the novelty of the theoretical contributions is limited, and the analysis does not convincingly establish that HDRQ can resolve merging issues beyond what standard quantization errors would introduce.

---

> ### Author Rebuttal · Authors · 2025-04-01
>
> First and foremost, we sincerely appreciate your thoughtful comments.
>
> ## Concern 1 : Limited merging setting
>
> We emphasize that our work deliberately targets the most challenging scenario for enabling domain adaptation on resource-constrained edge devices. Specifically, we design our setting to be demanding in the following four aspects:
> 1. Edge devices lack sufficient resources for fine-tuning.
> 2. State-of-the-art PTQ algorithms are also too costly for such devices.
> 3. To minimize communication costs, only quantized models can be deployed.
> 4. Merging must remain lightweight.
>
> While exploring boundary-breaking methods is important, we also believe addressing practical and challenging real-world deployment scenarios is equally critical. From this view, complex, data-intensive merging methods fall outside our scope. Given the strict resource constraints and latency sensitivity of edge environments, merging must remain lightweight.
>
> With this in mind, we adopt the interpolation-based merging strategy proposed in [1]. Notably, our baseline outperforms prior MTDA techniques, demonstrating its strength as a representative of state-of-the-art trends in model merging.
>
> [1] Wenyi Li, Huan-ang Gao et al, “Training-Free Model Merging for Multi-target Domain Adaptation”, ECCV 2024
>
> ## Concern 2 : Attribution of accuracy degradation
>
> As shown in Table 1 (a) and (b), quantization degrades each model's quality and negatively impacts merged performance. However, when quantization is done with merging in mind, quality is better preserved. In Table 1 (a), HDRQ matches QDrop in sole quantization but significantly outperforms it in harmonic mean(1.69 mIoU / 4.14 mIoU, before/after correction).  This highlights the importance of merging-aware quantization.
>
> ## Concern 3 : Baseline performance
>
> We acknowledge that HDRQ initially underperformed QDrop and have devoted significant effort to address this. We identified a minor bug in our implementation of HDRQ—the quantization step size was not updated during training—leading to negatively biased results.
>
> Corrected results are shown below:
>
> |    |  | W8A8    | W8A4    | W4A8    | W4A4    | W3A3    |
> |:---:|:----:|:----:|:---:|:---:|:----:|:---:|
> | **R → ACP**  | QDrop | **67.91** | **67.4**  | 64.85  | 66.26  | 62.99  |
> |       | HDRQ  | 67.46 | 66.71 | **66.75** | **66.41** | **64.7**  |
> | **A → RCP**  | QDrop | **68.75** | **68.24** | 66.83  | 66.04  | 64.22  |
> |        | HDRQ  | 68.1  | 68.15 | **67.8**  | **67.58** | **65.29** |
> | **C → RAP**  | QDrop | **74.79** | **74.58** | 73.81  | 73.25  | 71.01  |
> |        | HDRQ  | 74.58 | 73.7  | **74.26** | **73.57** | **71.63** |
> | **P → RAC**  | QDrop | **64.8**  | **64.52** | 62.52  | **63.22**  | 61.24  |
> |         | HDRQ  | 64.51 | 64.09 | **63.93** | 63.19  | **61.55** |
>
>
> HDRQ generally outperforms QDrop at lower bit-widths of weight (W4A8, W4A4, W3A3). In 11 wins, HDRQ’s average margin is 0.95—nearly double the 0.41 margin across 9 losses. These results demonstrate  HDRQ’s consistent superiority over QDrop in many scenarios. We will revise all results in the final paper. Thank you again for the feedback led to this correction.
>
>
> ## Concern 4 : Comparison with SoTA PTQ method
>
> While we recognize the importance of comparing with recent methods, we note that BRECQ and QDrop remain strong and reproducible baselines despite earlier publication. Still, we agree on the importance of comparison with recent methods and conducted further experiments.
>
> Specifically, we included FlexRound as a recent baseline, but excluded PTMQ due to its multi-bit focus and weak single-bit performance. We also tried to reproduce Shang et al.’s method, but the lack of official code and limited time prevented inclusion.
>
> We validated implementation using 4W4A quantization on ResNet-18 with the official PyTorch checkpoint (top-1: 69.76%). FlexRound achieved 67.63%, and QDrop 67.87%, consistent with original reports (~0.2%p difference).
>
> Then we compared our method and FlexRound on semantic segmentation under 4W4A setting:
>
> | | FlexRound | HDRQ  |
> |:------:|:-----:|:----:|
> | **Sole - CS**  |   60.01   | 59.76 |
> | **Sole - IDD**  |   48.94   | 48.51 |
> | **Merged - HMean** |   46.44   | **46.73** |
>
> While FlexRound is slightly better standalone, HDRQ achieves a higher harmonic mean-highlighting the advantage of merging-aware quantization.
>
> ## Concern 5 : Model merging strategy
>
> As detailed in our motivation, our goal is real-time model merging on edge devices. While advanced methods could improve merging quality, edge devices often lack resources to handle complex techniques. Thus, we adopt interpolation-based merging as our default, given the limited research on quantization-merging interplay. We hope this work offers valuable insights in this direction.
>
> We also acknowledge the missing key references and will include the suggested citations in the final version.

---

> > ### Comment · Reviewer_zjyC · 2025-04-01
> >
> > Thanks for your response. The results are well-received on my end. I raise my score. But since the benefit from the method is kind of marginal, I will keep my rating but will not fight for its acceptance.

---

> > > ### Author Response · Authors · 2025-04-04
> > >
> > > Dear Reviewer zjyC,
> > >
> > > We sincerely appreciate your thoughtful feedback and constructive comments. Thank you for the time and effort you dedicated to reviewing our work.
> > >
> > > Although our rebuttal may not have fully addressed all of your concerns to the extent of earning a more positive assessment, we greatly value your insights and will carefully incorporate them as we refine our paper for the final version.
> > >
> > > Once again, we are truly grateful for your effort and contributions to the review process.

---

### Official Review · Reviewer_5KmE · 2025-03-06

**Overall Recommendation:** 3

**Summary:**

This paper proposes a merge-friendly post-training quantization method for multi-target domain adaptation. It considers quantization and domain adaptation simultaneously. They propose HDRQ (Hessian and Distance Regularization Quantization), a post-training quantization method to preserve merging compatibility in multi-target domain adaptation. Hessian Regularization controls sensitivity to perturbations, while distance regularization reduces weight divergence among models. A noise-sampling-based rounding mechanism is proposed to alleviate rounding ambiguity. Experiment results prove the effectiveness of the proposed method.

**Claims And Evidence:**

I think the claims in this paper are supported by convincing evidence.

**Essential References Not Discussed:**

N/A

**Experimental Designs Or Analyses:**

I think the model design is reasonable and supported by theoretical claims. The experiment designs and analyses are also reasonable and clear. My concern is about the experiment results on more complicated tasks.

**Methods And Evaluation Criteria:**

I think the proposed method is reasonable. The evaluation criteria are also reasonable, but I think the authors can include more complicated datasets or tasks, eg. for image classification, I am curious about the method performance on larger datasets such as DomainNet.

**Other Comments Or Suggestions:**

N/A

**Other Strengths And Weaknesses:**

Strength:
1. The paper is overall well-written, with a very clear organization.
2. The theoretical insights of this paper are demonstrated clearly, and it naturally supports the model design.
3. The proposed method is reasonable and interesting to me.

Weakness:
1. As mentioned, I think the authors can provide more results on more complicated datasets or tasks.

**Questions For Authors:**

See weaknesses.
## After Rebuttal
After reading the rebuttal and other reviews, I still tend to accept this paper.

**Relation To Broader Scientific Literature:**

I think proposing merge-friendly quantization method for a specific downstream task is beneficial to the community and application of deep learning.

**Theoretical Claims:**

I have checked the theoretical claims in Chapter 3. They are clear and can support the method design.

---

> ### Author Rebuttal · Authors · 2025-04-01
>
> Thank you for your insightful review and constructive feedback on our submission.
>
> ## Weakness 1 : Complicated data and tasks
>
> We agree that experiments on more complicated tasks are truly helpful. We appreciate your suggestion and will consider additional experimental results.
>
> Due to the limited resource and time, we are not able to produce full experimental results for domainnet, but we here provide partial merging results of 3W3A quantized DA models from Real → Clipart, Sketch, Painting:
>
> |                                  | BRECQ | QDROP | HDRQ  |
> |:--------------------------------:|:-----:|:-----:|:-----:|
> | **Harmonic Mean (R → C, S, P)** |  34.07 |  41.51 | **42.08** |
>
> In this moment, we hypothesize HDRQ to work well with larger domain shift, since domain adapted weights are expected to still remain in single basin, where our theoretical analysis are valid.  We expect our method to either surpass baseline methods or show comparable performance in other settings, as observed in the Office-Home dataset. We will include the full experimental results in the camera-ready version. Thank you for your invaluable suggestion.

---

> > ### Comment · Reviewer_5KmE · 2025-04-05
> >
> > After reading the rebuttal, I still tend to accept this paper.

---

> > > ### Author Response · Authors · 2025-04-05
> > >
> > > Dear reviewer 5KmE,
> > >
> > > Thank you for your thoughtful review and for sharing your perspective on our work. We appreciate that you remain inclined to accept the paper, and we're grateful for the insights you provided.
> > >
> > > Thank you again for your contribution to the review process.

---

### Official Review · Reviewer_PqA3 · 2025-03-19

**Overall Recommendation:** 3

**Summary:**

This paper introduces HDRQ (Hessian and Distance Regularizing Quantization), a novel post-training quantization (PTQ) method designed to improve merge-friendly quantization for multi-target domain adaptation. Model merging has been shown to be an effective way to adapt models across multiple target domains, but quantization introduces discretization effects that degrade merging performance. This work systematically analyzes the impact of quantization on model merging through the lens of error barriers, showing that quantization-induced perturbations disrupt weight alignment, leading to suboptimal merging.

**Claims And Evidence:**

The paper presents a well-structured argument for HDRQ, and its claims are largely supported by theoretical analysis and empirical results. The paper provides a theoretical analysis using the error barrier concept, showing that quantization noise misaligns weights and increases merging degradation. Experiments on both semantic segmentation and classification tasks demonstrate that HDRQ achieves comparable or superior accuracy to standard PTQ methods in single-model settings while significantly improving merged model accuracy.

**Essential References Not Discussed:**

No

**Experimental Designs Or Analyses:**

I examined the experimental design and analysis in the paper, particularly focusing on the validity of the experimental setup, evaluation metrics, and statistical rigor. The use of GTA → Cityscapes, Indian Driving Dataset for segmentation and Office-Home dataset for classification aligns well with multi-target domain adaptation (MTDA). Quantization is applied after domain adaptation but before merging, which mirrors a realistic deployment scenario where models are adapted to different target domains and then quantized for efficiency. The paper incrementally removes components (noise-based quantization, distance regularization) and shows their impact on performance.

**Methods And Evaluation Criteria:**

The proposed methods and evaluation criteria generally make sense for the problem of post-training quantization (PTQ) in multi-target domain adaptation (MTDA) via model merging.  The use of semantic segmentation datasets (GTA → Cityscapes, Indian Driving Dataset) is appropriate since segmentation is a resource-intensive task where quantization is highly relevant. The Office-Home dataset (Real, Art, Clipart, Product) is a well-established benchmark in domain adaptation, making it a reasonable choice for testing multi-target domain adaptation in image classification. The paper correctly evaluates the harmonic mean of accuracy across target domains, which is a standard metric for multi-target domain adaptation to ensure balanced performance across different domains.

**Other Comments Or Suggestions:**

No

**Other Strengths And Weaknesses:**

Strengths: 1. The paper introduces HDRQ, a merge-friendly post-training quantization (PTQ) method, which is a novel contribution. While quantization and model merging have been studied separately, this work is the first to systematically analyze how quantization affects merging and propose a quantization method explicitly designed for multi-target domain adaptation (MTDA) via model merging. 2. The use of error barriers to analyze quantization effects on merging adds a theoretical foundation to a practical problem, which is an underexplored area in PTQ. 3. It enhances the practicality of training-free multi-target domain adaptation, making real-time adaptive AI more feasible without retraining.

Weakness: 1. The paper claims HDRQ is the first to study quantization-aware model merging, but it does not compare against quantization-aware fine-tuning methods (e.g., QAT approaches that may mitigate merging issues). 2. Since edge deployment efficiency is a key motivation, including latency/runtime comparisons would improve the practical significance. 3. in highly heterogeneous domains, where domain shifts are large, does HDRQ still work? A failure case analysis would help clarify the method’s limitations.

**Questions For Authors:**

1. In Section 3.2, you state that HDRQ flattens the loss surface using Hessian regularization, and Figure 2 visually supports this claim. However, you do not provide a quantitative metric for Hessian smoothness. Could you report the Hessian eigenvalue distribution or a sharpness measure (e.g., Spectral Norm of Hessian, Fisher Information Matrix, or Sharpness-Aware Minimization (SAM) curvature metrics) to validate this claim?

2. The results in Tables 1 and 2 report performance improvements, particularly in low-bit settings (e.g., W4A4, where HDRQ achieves +1.69 mIoU over QDrop). However, standard deviations and statistical significance tests are not reported. Could you provide standard deviations over multiple runs and conduct a paired significance test (e.g., t-test or Wilcoxon signed-rank test) to ensure that the improvements are statistically meaningful?

3. HDRQ is tested on ResNet-based architectures (ResNet-50, ResNet-101), but modern resource-efficient models (e.g., Vision Transformers, MobileNets, EfficientNet) are widely used in real-world low-power applications. Have you tested HDRQ on non-ResNet architectures? If not, do you anticipate challenges in applying HDRQ to architectures with different weight distributions (e.g., ViTs)?

4. HDRQ introduces Hessian-based smoothing, distance regularization, and noise-sampling-based rounding. How does this impact quantization time, inference speed, and memory overhead compared to standard PTQ methods (e.g., QDrop, BRECQ)? Can you provide runtime comparisons (in milliseconds per sample) and memory usage measurements?

5. HDRQ includes a noise-sampling-based rounding technique to improve merging stability, but no direct ablation is provided for this component. Could you report results comparing HDRQ with and without noise-based rounding to show its direct impact on merging performance?

**Relation To Broader Scientific Literature:**

The key contributions of this paper, HDRQ (Hessian and Distance Regularizing Quantization), are closely related to several existing areas in the broader scientific literature, including model merging, post-training quantization (PTQ), multi-target domain adaptation (MTDA), and loss landscape analysis.

**Theoretical Claims:**

Yes, I checked the proof and found no issues.

---

> ### Author Rebuttal · Authors · 2025-04-01
>
> Thank you for your valuable feedback. We've covered your key points and clarified our work.
> ## Weakness 1 : Comparison with QAT
> Quantization-aware training (QAT) typically yields higher quantization quality compared to post-training quantization (PTQ), but it demands full access to training data and significantly more computational resources. In our setting, which centers on per-device domain adaptation, the available target data is often too limited to support end-to-end fine-tuning without risking overfitting. To ensure practical applicability, we focus on PTQ and compare our approach against baseline PTQ methods.
> ## Weakness 2 : latency/runtime comparisons
> In our setup, fine-tuning and PTQ are done on a server, while only model merging and inference run on the edge device. Since the quantized models retain the same bit configuration and granularity, their latency and runtime remain unchanged after merging.
>
> As a result, the only remaining computational component on the edge device is the model merging process. To clarify its cost, we measured the time on a Raspberry Pi 5 using MobileNet-V2. The process took 35.53 seconds with 100 noise sampling iterations on an ARM Cortex-A76 CPU—about 3.60 seconds per 10 iterations—showing that merging is feasible even on low-power devices.
> ## Weakness 3 : Highly heterogenous domains
> Domain adaptation typically uses a lower learning rate and fewer epochs than training from scratch, so we assume that the resulting weight deviations are not large enough to escape a single basin—even under significant domain shifts.
>
> However, we acknowledge that if such a case occurs, additional treatments may be required to maintain the quality of quantized networks.
>
> We conducted a partial experiment on DomainNet (Real, Clipart, Sketch, Painting), a more complex dataset, and observed consistent improvements over baselines. Please refer to our response to Reviewer 5KmE’s Weakness 1.
> ## Question 1 : Sharpness measure
> As a measure of sharpness, we report the Hessian trace of the Real→Clipart domain-adapted model quantized to W4A8 on the Office-Home benchmark. The table below shows that HDRQ exhibits a lower Hessian trace, indicating better flatness:
> | |QDrop|HDRQ|
> |:-:|:-:|:-:|
> | **Hessian Trace** |42171|38348|
> ## Question 2 : Multiple runs
> We conducted five additional experiments with different seeds and report the mean and standard deviation:
> | |QDrop|HDRQ|
> |:-:|:-:|:-:|
> | **4W4A results (Harmonic mean)**|41.97 (±1.27)|46.14 (±0.74)|
>
> We would like to sincerely apologize for an error identified during the revision process. Specifically, we discovered that in our earlier experiments, the quantization scale in our method was not properly updated, which led to the reporting of degraded accuracy. In the updated results presented in this table, we have corrected this implementation bug. As a result, the overall accuracy has improved significantly. While we deeply regret this oversight, we also hope for your understanding that this issue introduced a negative bias in evaluating the performance of HDRQ.
> ## Question 3 : Other architectures
> Since our analysis is agnostic to model architecture, we expect it to generalize well to vision-oriented models. However, architectures like transformers—which often exhibit unique characteristics such as outlier channels—may require additional techniques to preserve quantization quality. That said, existing baseline methods are also likely to experience similar degradation in such cases. We plan to include results on ViTs and other architectures in the final version.
> ## Question 4 : Runtime, memory comparison
> Here is a table comparing the maximum memory consumption and quantization time of baseline methods and ours:
> | |BRECQ|QDROP|HDRQ|
> |:-:|:-:|:-:|:-:|
> | **Peak Memory Consumption (MB)** |9394|14643|14586|
> | **Quantization Time (hours)** |2.45|2.12|2.5|
>
> As shown, quantization time slightly increases over QDrop due to noise sampling at each iteration. Theoretically, memory usage may also rise slightly, as sampled noise must be stored for each gradient step. However, actual measurements via torch.cuda.max memory allocated show similar memory consumption due to PyTorch's caching mechanism. From the result, this allows us to assume that the memory overhead of HDRQ is minimal.
>
> In summary, these increases in memory consumption and overall time are minimal and represent an affordable cost.
>
> The inference speed of models quantized with each method remains the same, as the quantized models have the same bit configuration and quantization granularity.
> ## Question 5 : Ablation of noise-sampling-based rounding technique
> The ablation study is provided in Figure 3 of the main paper, where "Naive" denotes merging without advanced sampling. For the merging of ambiguous cases with multiple valid quantization levels, the Naive method assigns one at random. Advanced sampling filters out low-quality samples, improving average performance and reducing distribution tail size.

---

### Official Review · Reviewer_UftN · 2025-03-20

**Overall Recommendation:** 3

**Summary:**

This paper investigates the impact of quantization on model merging in multi-target domain adaptation. The key insight is that prior approaches, which quantize the model before merging, degrade merging quality. To address this, the paper introduces HDRQ, a merge-friendly quantization method that incorporates two regularization terms: one to control sensitivity to perturbations and another to minimize weight divergence between models. The authors evaluate the proposed method on two tasks: semantic segmentation and image classification. Experimental results demonstrate that the proposed method HDRQ enhances model merging performance across various adaptation settings.

**Claims And Evidence:**

Yes. It looks so. The insight that quantization induces misalignment and affects the model merging process makes sense to me.

**Essential References Not Discussed:**

- The quantization method relies solely on vanilla round-to-nearest. A discussion on more advanced methods, such as AWQ, would be beneficial.

**Experimental Designs Or Analyses:**

The experimental design is well-structured and supports the main claim of this work.

**Methods And Evaluation Criteria:**

The proposed method appears reasonable; however, since the quantization approach is limited to vanilla round-to-nearest, it remains unclear whether more advanced quantization methods could further improve its effectiveness.

**Other Comments Or Suggestions:**

- Some implementation details of quantization are unclear, such as the quantization granularity and whether the method is based on RTE. Additionally, how would more advanced approaches like AWQ, SmoothQuant, SpinQuant, or GPTQ compare to the proposed method?

**Other Strengths And Weaknesses:**

- The main motivation is unclear. If the goal is to obtain a quantized merged model for efficient deployment, why not merge two unquantized models first and then apply post-training quantization? Could the authors elaborate on the practical applications to provide a clearer motivation?

- In Table 3, the improvements from distance regularization are not significant. How do the results vary with changes in the hyperparameter λ?

- In Table 1 (W4A4), the best results come from QDrop, yet the highlighted number corresponds to the proposed method. Could the authors clarify this discrepancy?

**Questions For Authors:**

N/A

**Relation To Broader Scientific Literature:**

- This work analyzes the interplay between quantization and model merging, offering a valuable contribution to the model merging literature.

**Theoretical Claims:**

This theoretical claim in Section 3.1 makes sense.

---

> ### Author Rebuttal · Authors · 2025-04-01
>
> We sincerely thank you for thoughtful feedback and constructive comments.
>
> ## Weakness 1 : Main motivation
> To begin with, we would like to emphasize that our work deliberately addresses the most challenging scenario for enabling domain adaptation on resource-constrained edge devices. Specifically, we design our setting to be highly demanding in the following four aspects:
> 1.  The edge device lacks sufficient resources to support fine-tuning operations.
> 2.  State-of-the-art PTQ algorithms are also too computationally expensive for such devices.
> 3.  To minimize communication costs, only quantized models can be deployed.
> 4. Model merging operations must remain lightweight.
>
> From this perspective, as some reviewers pointed out, an alternative approach—merging two unquantized models first and then applying PTQ—could be a viable solution. However, this approach becomes less practical when progressive adaptation is considered.
> In the case of progressive adaptation, additional merging must be performed on top of the previously merged weights. In our proposed scenario, the target model is expected to accumulate updated weights locally over multiple rounds. Since the server discards the personalized model after each fine-tuning round, this approach imposes a strong constraint. Nonetheless, it provides significant security advantages, as personalized models are not stored centrally. By contrast, in the alternative approach, the server must maintain a separate customized model for each device, which leads to increased storage overhead and raises potential concerns regarding information leakage. Please note that our proposed scheme works really well on our challenging scenario, showing the outstanding performance of the proposed idea.
>
>
>
> ## Weakness 2 : Results vary with changes in the hyperparameter λ
> Here, we present the results of model merging with varying values of $\lambda$. As shown in the table below, increasing $\lambda$ initially improves performance due to the regularization effect. However, beyond a certain point, excessive regularization starts to harm the quality of reconstruction during the quantization process, leading to accuracy degradation. Based on empirical observations, we selected $\lambda = 5 \times 10^{-2}$ as a balanced choice that consistently performs well across different settings. We adopt this value uniformly in all experiments throughout our paper.
>
> | λ   | 5e-2  | 7e-2  | 9e-2  | 1.1e-1 | 1.5e-1 |
> |---------|-------|-------|-------|--------|--------|
> | Harmonic Mean of Accuracy (%) | 66.75 | 66.78 | 66.84 | 66.98  | 66.86  |
>
>
> ## Weakness 3 : Discrepancy in highlighted number
> Sorry for the confusion, and thank you for pointing that out. As you correctly noted—particularly in Table 1(a)—the performance of QDrop should be more clearly emphasized. Our main intention was to highlight the quality improvement achieved after model merging, as demonstrated in Table 1(b). To avoid misunderstanding, we will revise Table 1(a) to clearly indicate that QDrop is the best-performing method for standalone quantization.
>
>
> ## Other comments : Comparison with AWQ, SmoothQuant, SpinQuant, or GPTQ
> In our experiments, we apply per-tensor, round-to-nearest quantization with truncation. Notably, the truncation range is learnable, following the formulation used in QDrop.
>
> AWQ, SmoothQuant, SpinQuant, and GPTQ are designed specifically for LLMs, and thus differ fundamentally from gradient-based methods like QDrop. These algorithms perform progressive, layer-wise quantization without iterative updates to reduce the computational burden—an essential property given the vast number of parameters in LLMs. However, in the case of vision models, QDrop often outperforms these LLM-oriented techniques. For this reason, our primary focus is on vision-oriented PTQ approaches.
>
> That said, we agree it is also valuable to assess the effectiveness of our proposed method under more advanced PTQ schemes, even if they were originally developed for LLMs. To this end, we conducted 4W4A quantization experiments on semantic segmentation using SmoothQuant, and compared our method against relevant baselines:
>
> |  | SmoothQuant | Qdrop  | HDRQ  |
> |:------:|:----------:|:------:|:-----:|
> | **6W6A** |  44.69  |  52.62 | **53.95** |
> | **4W4A** |  5.43   |  42.5  | **46.21** |
>
> As shown in the table, ours performs significantly better than SmoothQuant, especially in the low-bit W4A4 case. We will update the results in the camera-ready version.

---

> > ### Comment · Reviewer_UftN · 2025-04-04
> >
> > Thanks for the detailed response. I have increased my rating, but will not advocate strongly for the acceptance.

---

> > > ### Author Response · Authors · 2025-04-04
> > >
> > > Dear reviewer UftN,
> > >
> > > We sincerely appreciate the time and effort you dedicated to reviewing our work. Your thoughtful feedback and reassessment are truly meaningful to us, and we are grateful for the valuable insights you have shared.
> > >
> > > Thank you once again for your contribution to the review process.

---

### Decision · Program_Chairs · 2025-05-01

**Decision:**

Accept (poster)

**Comment:**

This paper proposes ​​HDRQ​​, a merge-friendly post-training quantization method for multi-target domain adaptation, addressing how standard quantization disrupts weight alignment during model merging. While reviewers initially raised concerns about motivation clarity, baseline comparisons, and marginal gains over QDrop, the authors clarified the edge-device use case, corrected implementation bugs (improving results), and added comparisons with recent methods eg. FlexRound. Allthough benefits remain modest and advanced merging techniques were untested, the paper’s theoretical contribution and practical relevance for resource-constrained scenarios justify acceptance. Thus the AC recommends acceptance. Congratulations to the authors!